# How Perceived Positive Parenting Style Protects Against Academic Procrastination in Children: The Mediating Roles of Emotional Resilience and School Emotional Engagement

**DOI:** 10.3390/bs15070890

**Published:** 2025-06-30

**Authors:** Junfeng Wei, Wenhao Gu, He Xiao, Yangang Nie

**Affiliations:** 1Research Center of Adolescent Psychology and Behavior, School of Education, Guangzhou University, Guangzhou 510006, China; 2112308029@e.gzhu.edu.cn (J.W.); gwhpsy@e.gzhu.edu.cn (W.G.); 2Center for Studies of Psychological Application, South China Normal University, Guangzhou 510635, China

**Keywords:** parenting style, emotional resilience, academic procrastination, school emotional engagement, children

## Abstract

Academic procrastination is a prevalent issue among children, often linked to poorer developmental outcomes. Prior research has uncovered cognitive, motivational, and dispositional antecedents of procrastination, yet its emotional correlates remain underexplored. Given the central role parenting plays in children’s emotional development, examining the emotional pathways through which parenting influences academic procrastination may deepen the understanding of emotional processes underlying academic development. Grounded in the Broaden-and-Build Theory of Positive Emotions, the present study examined the extent to which emotional resilience and school emotional engagement mediate, both individually and sequentially, the relationship between perceived positive parenting styles and academic procrastination in children. Drawing on three waves of data, this study employed structural equation modeling to assess a chain mediation model. The sample comprised 728 primary school students (Mage = 9.84, SD = 0.77, 49.22% female, range = 8 to 12 years) from Guangzhou, China. Participants completed the assessment at three time points (i.e., November 2021, May 2022, May 2023). The results reveal that perceived positive parenting styles significantly predict lower levels of academic procrastination. Both emotional resilience and school emotional engagement independently mediate the relationship between positive parenting style and academic procrastination. Moreover, this relationship is sequentially mediated by emotional resilience and school emotional engagement. While the mediation effect sizes were relatively small, the study identifies the emotional mechanism through which the perceived positive parenting influences children’s academic procrastination. The preliminary findings contribute to a richer understanding of the emotional underpinnings of academic procrastination and propose potential directions for future research and intervention.

## 1. Introduction

Striving for academic success constitutes a central developmental task for children in China. Yet, this pursuit is often hindered by various obstacles, with academic procrastination being one of them. While much of the existing research has centered on university students, with prevalence estimates ranging from 10% to 70% ([78]), academic procrastination is not limited to older students ([31]). [82] ([82]) contended that the severity of procrastination in elementary school students is similar to that in secondary school students and even more pronounced than in university students. Although a consistent prevalence rate has not been established yet, academic procrastination already becomes a salient concern among the youth as it not only interferes with the accomplishment of desired academic performance ([37]) but also impairs their emotional and psychological well-being ([18]; [60]; [61]).

Traditionally, procrastination has been viewed as an avoidance strategy in response to unpleasant tasks and has been linked to self-regulation failure ([77]), fear of failure ([69]), and poor action control ([72]). Although emotions have not been the primary focus in earlier research, they are increasingly recognized as integral to the process of procrastination. Specifically, aversive affective experiences often precede procrastinating behaviors ([77]); paradoxically, procrastination is frequently employed to mitigate emotional distress, particularly in response to demanding tasks. This dynamic could create a self-reinforcing cycle of negative emotion and avoidance ([71]; [61]).

While most research links procrastination to negative affect and avoidance, the Broaden-and-Build Theory offers a complementary perspective: positive emotions may help individuals break free from this negative cycle by building durable psychological resources ([27]). Among these resources, emotional resilience, defined as the capacity to sustain positive emotions and recover from adversity ([20]), emerges as particularly relevant to the academic context. Students with greater emotional resilience are more capable of managing stress, regulating negative emotions, and maintaining engagement when faced with challenging academic tasks, thereby reducing their vulnerability to procrastination ([59]; [63]). Thus, fostering positive emotions and cultivating emotional resilience may represent a promising approach to disrupting the self-reinforcing cycle of procrastination and distressing emotions.

Besides emotional capacities at the individual level, emotionally relevant experiences within the school environment also warrant attention due to their proximal influence on academic behaviors ([10]). School emotional engagement—students’ emotional connection to teachers, peers, and the broader school context—has been linked to stronger academic performance and well-being ([51]; [92]; [56]). Supportive relationships at school counterbalance the impacts of academic stressors and help sustain students’ commitment, thereby reducing the likelihood of procrastination ([68]; [85]).

Parenting styles are vital in shaping children’s emotional resilience and school engagement. Positive parenting, characterized by warmth, responsiveness, and autonomy support, has been shown to foster resilience ([17]; [29]) and enhance school engagement ([80]; [79]). Importantly, it is not only parenting behaviors themselves but also children’s perceptions—how they experience and construe parenting—that influence their development. Despite the evidence substantiating the impacts of parenting styles on academic procrastination ([75]; [81]; [97]), the mechanisms involving children’s emotional attributes and relational experiences are insufficiently understood. Hence, we propose a chain mediation model to explore how children’s perceptions of parenting styles influence academic procrastination through emotional resilience and school emotional engagement individually and sequentially.

### 1.1. Parenting Styles and Academic Procrastination

[5] ([5], [6]) conceptualized parenting styles as a constellation of integrated parenting practices. Based on patterns in how parental control and responsiveness are combined, she identified three classical parenting styles: authoritative, authoritarian, and permissive. In Western contexts, authoritarian parenting—characterized by high control and low responsiveness—is often associated with maladaptive outcomes such as poor emotional adjustment and strained parent–child relationships ([43]). However, in Chinese society, which is rooted in Confucianism and collectivism, parent–child relationships have traditionally been hierarchical, with cultural norms emphasizing children’s obedience to parental authority ([15]). In this cultural milieu, authoritarian parenting was widely practiced; furthermore, it was historically linked to reduced behavioral problems and even positive developmental outcomes in Chinese children ([14]; [90]).

As modern educational values such as openness, mutual respect, and emotional communication gain wider acceptance in China, contemporary parent–child relationships have gradually become more egalitarian ([93]). This cultural shift has been accompanied by a growing preference for authoritative parenting, which balances discipline with emotional warmth and support. Recent studies indicate that, among Chinese children, authoritative parenting is associated with better academic and socioemotional outcomes compared with authoritarian parenting ([15]; [49]).

As family science evolved, the conceptual scope of parenting style has been expanded beyond the traditional typology based on responsiveness and control; it has come to be understood more broadly as the overall pattern of parents’ emotional expressions and behavioral interactions in childrearing ([19]). Under this conceptual framework, parenting styles could be categorized as either positive or negative. Positive parenting styles, characterized by compassion, emotional responsiveness, and autonomy support, are associated with adaptive developmental outcomes in children ([64]). In contrast, negative parenting styles, typically involving rejection, dismissiveness, harsh discipline, and rigid control, have been linked to socioemotional maladjustment and poorer mental health ([81]).

The literature has illustrated that children under positive parenting styles tend to display lower levels of academic procrastination as the associated parenting practices foster a range of intrapersonal capacities in children, such as self-esteem, self-efficacy, time management skills, and conscientiousness ([75]; [81]; [97]). These attributes enhance a child’s confidence, motivation, organization, and persistence, all of which help lessen procrastination ([4]; [36]; [46]; [76]; [97]). However, these findings are more concerned about cognitive, motivational, and dispositional pathways; less is known about whether children’s emotional functioning plays a role. Given the persistent emphasis Chinese parents place on academic achievement ([51]), examining emotional mechanisms could contribute to a more nuanced understanding of how parenting influences students’ academic behaviors, including procrastination.

### 1.2. The Mediating Effect of Emotional Resilience

Emotional resilience, theorized as the ability to generate and sustain positive emotions in response to adverse stimuli or events ([20]), has emerged as a potential mediator. Individuals with high emotional resilience are more likely to adopt effective emotional regulation strategies to cope with academic stressors ([83]). Meta-analytic evidence indicates that individuals with higher emotional resilience are better able to manage negative emotions ([63]). Prior research has found that individuals with greater emotional regulation capability are less likely to engage in procrastination ([24]; [94]). According to the Broaden-and-Build Theory ([27]), positive emotions expand the momentary thought-action repertoire, enabling individuals to move beyond the established behavior patterns and achieve more long-term adaptive outcomes. Empirical findings also show that positive emotions enhance learning motivation, self-efficacy, and academic engagement, which contribute to improved academic performance ([48]; [58]; [59]). In the face of academic challenges, emotionally resilient students are possibly better at navigating emotional discomfort by regulating aversive emotions and retaining positive ones, rendering themselves less vulnerable to academic procrastination.

Positive parenting practices offer warmth, security, and positive emotional experiences ([8]). These foundational elements not only help allay emotional strain ([89]) but also undergird the development of emotional resilience in children ([17]; [29]). For instance, parents who validate and acknowledge their children’s emotions and actively model constructive coping strategies serve as important socialization agents for the development of children’s emotion regulation skills. Systematic evidence shows that consistent emotional support and open parent–child communication not only promote children’s effective stress management but also foster greater emotional competence and resilience ([26]). Thus, emotional resilience may function as a key mediator linking positive parenting practices to reduced academic procrastination in children.

### 1.3. The Mediating Effect of School Emotional Engagement

School emotional engagement refers to students’ emotional connection with school and their identification with academic activities ([41]). A strong emotional bond with the educational environment can motivate students to participate more fully in learning and reduce their reliance on procrastination as an avoidance strategy ([21]; [23]). Supportive relationships with teachers and peers buffer the negative emotions associated with academic stress ([68]) and facilitate effective problem-solving ([85]). A recent study identified a negative relationship between school emotional engagement and academic procrastination among primary school children, suggesting that emotional engagement could act as a protective mechanism ([82]).

A growing body of research indicates that parental involvement and support significantly predict students’ school emotional engagement, thereby enhancing their academic adjustment and success ([84]). Positive parenting characterized by autonomy support and warmth enables children to develop a sense of security and belonging that extends into the school context, fostering stronger emotional connections with their educational environment ([79]). This process further promotes the development of social-emotional competencies such as emotion regulation and positive peer interactions, which are foundational for deeper school emotional engagement ([80]). Recent evidence also demonstrates that positive family involvement enhances students’ emotional connection to school through the development of these social-emotional skills ([57]). Accordingly, the present study posits that school emotional engagement mediates the relationship between positive parenting and academic procrastination.

### 1.4. The Chain Mediating Effect of Emotional Resilience and School Emotional Engagement

While emotional resilience and school emotional engagement may independently mediate the parenting–procrastination link, evidence suggests they may operate jointly in a sequential pathway, where emotional resilience fosters school emotional engagement, which in turn reduces academic procrastination. According to Kumpfer’s resilience framework (2002), resilience extends beyond the recovery to include social reintegration, a process through which individuals reconnect with their social environments following adversities. The reintegration can be reflected by re-establishing meaningful relationships and actively participating in social, academic, or community settings.

Prior studies have found that academic resilience, defined as the capacity to persist through academic difficulties ([55]), is positively associated with school emotional engagement (e.g., [38]; [56]; [67]). While academic resilience and emotional resilience are distinct constructs, they share a common ground of making adaptive responses to hardships. Hence, it stands to reason that emotional resilience could similarly promote students’ school emotional engagement by enhancing their capacity to manage stress and maintain positive emotional bonds with peers and teachers, and stay engaged in educational setting ([22]), which may lower the risk of avoidance-oriented behaviors like procrastination. Based on empirical and theoretical evidence, emotional resilience and school emotional engagement may function in a sequential manner to elucidate the pathway through which parenting style affects academic procrastination.

### 1.5. Current Study

To further understand the emotional mechanisms by which perceived positive parenting style influences children’s academic procrastination, this study employed three waves of data, applied a chain mediation model ([32]) to explore the mediation role of emotional resilience and school emotional engagement, and proposed the following research questions:

1. Does the perceived positive parenting style directly predict academic procrastination?

2. Does the perceived positive parenting style indirectly predict academic procrastination via emotional resilience?

3. Does the perceived positive parenting style indirectly predict academic procrastination via school emotional engagement?

4. Does the perceived positive parenting style indirectly predict academic procrastination via emotional resilience first and then school emotional engagement?

The hypothetical model is presented in Figure 1.

## 2. Method

### 2.1. Participants

Participants were recruited using a cluster sampling method. The final sample consisted of 728 typically developing children (Mage = 9.84, SD = 0.77; 49.20% female) from two regular public primary schools in a major city in southern China. The demographic characteristics in this study are shown in Table 1. Of 803 children at T1, 787 participated in the assessment at T2 (attrition rate = 1.99%), and 728 participated in the assessment at T3 (attrition rate = 9.34%).

### 2.2. Measures

#### 2.2.1. Perceived Positive Parenting Style

Perception of positive parenting styles were assessed using the Chinese short version of the Egna Minnen av Barndoms Uppfostran scale (s-EMBU-C; [62]; [34]). This construct subsumes three dimensions: rejection (e.g., “Father’s/Mother’s punishments exceed what I deserve.”), emotional warmth (e.g., “Father/Mother praises me.”), and overprotection (e.g., “I feel that my father/mother interferes with anything I do”). Participants reported perceived parenting styles of both their father and mother separately. The scale consists of 21 items. Items are rated on a 4-point scale (from “1 = *never*” to “4 = *always*”). The Cronbach’s alpha coefficient for the overprotection subdimension was below 0.6 and, therefore, was excluded from further analysis. Following the approach of [35] ([35]), the items from the rejection subdimension were reverse-scored and combined with those from the emotional warmth subdimension to construct a composite indicator of positive parenting style. A higher total score indicates a higher level of positive parenting. We further conducted a confirmatory factor analysis (CFA) to examine the structural validity of this composite construct. The results indicated that the measurement of perceived positive parenting style demonstrated good structural validity (χ^2^/df = 3.59, CFI = 0.960, TLI = 0.941, RMSEA = 0.06). The Cronbach’s alpha for positive parenting style is 0.862.

#### 2.2.2. Emotional Resilience

The current research utilized the Emotional Resilience Scale developed by [95] ([95]). The scale was based on [20]’s ([20]) concept of emotional resilience and had been validated in the Chinese population. The scale contains 11 items, subsuming two dimensions: positive emotional capability (e.g., “When I am in a bad mood, I think of happy things”) and emotional recovery capability (e.g., “I can quickly adjust my negative emotions”). Items are rated on a 6-point scale (from “1 = *strongly disagree*” to “6 = *strongly agree*”). A higher total score indicates stronger perceived emotional resilience. The internal consistency of emotional resilience was 0.802 at T1 and 0.838 at T2. A CFA was conducted on the scale using the T2 data. The results indicated that the scale demonstrated good structural validity (χ^2^/df = 3.99, CFI = 0.973, TLI = 0.954, RMSEA = 0.064).

#### 2.2.3. School Emotional Engagement

School emotional engagement was measured using the School Liking and School Avoidance Questionnaire (SLAQ) developed by [42] ([42]). The participants completed the Chinese version of the SLAQ. This scale consists of 11 items, covering two dimensions: school liking, which reflects students’ enjoyment and positive attitude toward school (e.g., “I like school.”), and school avoidance, which refers to the extent to which students fail to derive enjoyment from school participation (e.g., “I think it’s better not to go to school.”). Items are rated on a 5-point scale (from “1 = *strongly disagree*” to “5 = *strongly agree*”). Notably, these two dimensions capture children’s emotional attitudes toward school rather than their commitment to school engagement. A high score represents a high level of school emotional engagement. This scale has been validated for measuring Chinese children’s school emotional engagement ([51]). The Cronbach’s alpha values of school emotional engagement were 0.864 at T1 and 0.867 at T2. A CFA was conducted on the scale using the T2 data. The results indicated that the scale demonstrated good structural validity (χ^2^/df = 3.59, CFI = 0.984, TLI = 0.958, RMSEA = 0.060).

#### 2.2.4. Academic Procrastination

The Chinese version of the Academic Procrastination Scale was adopted. It was translated and revised by [47] ([47]) based on [44]’s ([44]) Academic Procrastination Questionnaire. The scale has been validated among Chinese primary school students. The scale consists of 19 items (e.g., “I often delay tasks that must be completed for my studies.”). Items are rated on a 5-point scale (from “1 = *strongly disagree*” to “5 = *strongly agree*”). The Cronbach’s alpha values of academic procrastination were 0.801 at T1 and 0.887 at T3. A CFA was conducted on the scale using the T3 data. The results showed that the scale demonstrated good structural validity (χ^2^/df = 3.14, CFI = 0.946, TLI = 0.934, RMSEA = 0.047).

### 2.3. Procedure

All procedures involving human subjects were reviewed and approved by the research ethics committee of the first author’s affiliated institution. As primary school students are under the age of 18, they were instructed to take the informed consent form home for parental signature. Teachers also explained the study to parents through online communication prior to the assessment. Data collection was conducted in November 2021 (T1), May 2022 (T2), and May 2023 (T3). Due to pandemic control measures, data collection for T3, which was originally scheduled for November 2022, was postponed to May 2023. All measurements were conducted in person, with participants completing the questionnaires offline at school. In each classroom, two trained graduate students assisted the participants throughout the process, including explaining questionnaire items and ensuring standardized completion procedures.

### 2.4. Data Analysis

In addition to employing SPSS version 26.0 to derive descriptive statistics and conduct correlation analyses, Harman’s single-factor test was performed using the principal component analysis (PCA). All measurement items from the study were included in the analysis. Following standard practice, components with eigenvalues greater than 1 were extracted, and the threshold of 40% of variance explained by the first component was used to assess the presence of common method bias.

The structural equation model was tested using Mplus 8.0. Missing data were addressed using full information maximum likelihood (FIML) estimation. In the structural equation model, T1 parenting style serves as the independent variable, T2 emotional resilience and school emotional engagement sequentially function as mediating variables, and T3 academic procrastination acts as the dependent variable (Figure 1). The T1 data of the mediator and outcome variables were included in the structural equation modeling to control for baseline effects. The mediation effect was tested using the model constraint command. Model fit was evaluated using standard fit indices, including the comparative fit index (CFI), Tucker–Lewis index (TLI), root mean square error of approximation (RMSEA), and standardized root mean square residual (SRMR), with values of CFI and TLI > 0.90, RMSEA < 0.08, and SRMR < 0.08 considered acceptable. Bootstrapping was used to extract 95% confidence intervals (CIs) to test the significance of the path coefficients (5000 samples). Standardized indirect effects, along with their confidence intervals, were utilized to assess the significance and magnitude of the mediation effects. To further evaluate the robustness of these effects, a Monte Carlo simulation was conducted to perform the post hoc power analysis. A statistical power of 0.80 or higher was considered acceptable, indicating a sufficient probability of detecting the mediation effect if it truly exists.

## 3. Result

### 3.1. Preliminary Analysis

The Harman single-factor test was employed to evaluate the common method bias. The analysis revealed that 25 factors had eigenvalues greater than 1, with the first factor accounting for 17.12% of the variance, which is well below the critical threshold of 40%. This result indicates the absence of significant common method bias. Descriptive statistics and the correlation matrix for all key variables and covariates are presented in Table 2. The relationships between key variables are in the expected direction.

### 3.2. Chain Mediation Model Testing

The chain mediation model, with the baseline levels of all key variables controlled for, demonstrated good model fit (χ^2^ = 79.297, df = 30, *p* < 0.001, RMSEA = 0.048, 95%CI [0.035, 0.060], CFI = 0.956, SRMR = 0.034). The results showed that the perceived positive parenting style at T1 negatively predicted academic procrastination at T3 (*β* = −0.110, *p* = 0.006) and positively predicted emotional resilience at T2 (*β* = 0.104, *p* = 0.002) but did not significantly predict school emotional engagement at T2 (*β* = 0.068, *p* = 0.054). Emotional resilience at T2 negatively predicted academic procrastination at T3 (*β* = −0.230, *p* < 0.001) and was significantly positively associated with school emotional engagement at T2 (*β* = 0.394, *p* < 0.001). School emotional engagement at T2 negatively predicted academic procrastination at T3 (*β* = −0.210, *p* < 0.001). The standardized structural equation modeling results are presented in Figure 2. The standardized total effect of perceived positive parenting style on academic procrastination was significant (*β* = −0.157, 95%CI = [−0.237, −0.084]). The results of the mediation analysis indicated that the perceived positive parenting style indirectly influenced children’s academic procrastination through three pathways: individually through emotional resilience (*β* = −0.024, 95%CI = [−0.045, −0.009]) and school emotional engagement (*β* = −0.014, 95%CI = [−0.033, −0.002]) and through emotional resilience and school emotional engagement sequentially (*β* = −0.009, 95%CI = [−0.017, −0.003]) (Table 3). It is noted that the non-significant effect of perceived positive parenting style on school emotional engagement does not determine the significance of the mediation effect of school emotional engagement in that a statistically significant indirect effect does not require each individual path to be significant ([32]).

The results of the Monte Carlo simulation indicate that the mediation effect for emotional resilience (*β* = −0.024, power = 0.915 > 0.8) and the chained mediation effect (*β* = −0.009, power = 0.893 > 0.8) are small but significant, and the results are reliable. However, the mediation effect for school emotional engagement (*β* = −0.014, power = 0.637 < 0.8), despite being statistically significant, should be interpreted with caution.

## 4. Discussion

With a sample of Chinese students, this study employed a chain mediation model to examine the extent to which emotional resilience and school emotional engagement mediate the relationship between perceived positive parenting style children’s academic procrastination. The findings illustrate that positive parenting style directly affects children’s academic procrastination (RQ #1). Perception of positive parenting style indirectly predicts academic procrastination through emotional resilience and school emotional engagement, respectively (RQ #2, RQ#3). Furthermore, emotional resilience and school emotional engagement sequentially mediate this relationship (RQ #4).

### 4.1. Positive Parenting Style and Academic Procrastination

The present study found a direct effect of positive parenting style on academic procrastination among Chinese children. This suggests that parents who provide more emotional warmth, responsiveness, and support, and engage in less rejection and invalidation, could help reduce their child’s tendency to procrastinate on academic tasks. This finding is consistent with prior research in both Western and Asian contexts ([4]; [19]; [49]; [45]; [75]). While ecological systems theory highlights the central role of parents in shaping child development ([10]), parenting practices are culturally embedded and may differ in expression across societies. It is, therefore, vital to situate the findings within the Chinese sociocultural context.

In Chinese society, parental warmth has been traditionally expressed through non-verbal behaviors such as instrumental support, guidance and even acts of sacrifice, rather than explicit displays of affection ([13]; [16]). However, this norm appears to be shifting. As younger generations of Chinese parents have been exposed to diverse perspectives of parenting, they gradually endorse and adopt positive parenting styles ([73]). Consequently, the direct expressions of emotional warmth, such as care, compassion, and autonomy support, are more common ([50]). These changes have been linked to improvements in children’s academic motivation and mindset ([65]; [70]), which may help explain the effectiveness of emotionally supportive parenting in reducing procrastination.

### 4.2. The Mediating Role of Emotional Resilience

This study found that the mediation effect of emotional resilience is significant but small in magnitude. Results derived from the Monte Carlo simulation confirm the reliability of this effect, suggesting that the associations, while limited in strength, are unlikely to be a statistical artifact. The finding suggests that emotionally supportive parenting may foster children’s emotional strength, which could further mitigate procrastination.

In line with previous research findings ([96]), students with higher emotional resilience are less likely to procrastinate. When facing academic stressors and experiencing negative emotions, emotionally resilient students are generally better able to manage negative emotional states ([28]; [9]) and restore positive emotions. This aligns with the Broaden-and-Build Theory of Positive Emotions ([27]), which posits that positive emotions expand individuals’ thought–action repertoires. The regaining of positive emotional states, due to the emotional resilience, may then enable students to think constructively and engage in active coping, planning, and positive reappraisal to render academic demands more manageable, instead of delaying academic tasks ([30]; [83]).

Resilience develops from the dynamic interplay between external risk and protective factors ([40]). For Chinese students, ongoing academic pressures could cause enduring emotional strains. Yet, consistently positive parenting can act as a buffer against these stressors. Over time, this balance between protective input and emotional risk may promote an upward spiral of emotional strength ([17]; [2]), gradually fostering children’s resilience in emotionally demanding contexts.

Positive parenting practices may promote emotional resilience not only through behavioral modeling and emotional support, but also through their influence on children’s physiological regulation systems. Secure attachment, which develops through positive parenting practices such as emotional responsiveness, has been associated with reduced activation of the hypothalamic–pituitary–adrenal (HPA) axis, a system that governs the body’s stress response ([11]; [33]). Lower HPA reactivity may help children manage stress more effectively and avoid becoming overwhelmed by negative emotions. Similarly, parental warmth has been shown to reduce activity in the amygdala, a brain region involved in emotional reactivity and threat detection, which contributes to greater emotional stability ([74]). Although these neurobiological processes were not directly assessed in the current study, such findings offer preliminary support for the potential benefits of positive parenting for children’s emotional functioning, including emotional resilience.

While this study demonstrates that emotional resilience could mediate the relationship between perceived positive parenting and academic procrastination, its small effect size implies the presence of other factors serving as more robust mechanisms. Future research could continue exploring a broader set of biological, mental, and contextual pathways.

### 4.3. The Mediating Role of School Emotional Engagement

The mediation effect of school emotional engagement was significant; however, the effect size was rather small, and the post hoc power analysis indicated its suboptimal statistical power, falling below the conventional threshold. As such, the individual mediation role of school emotional engagement should be interpreted cautiously and not viewed as a central finding of the study.

Prior studies indicate that positive parenting styles contribute to school emotional engagement ([53]; [88]). From the perspective of parental emotion socialization ([25]), emotionally supportive parenting, which represents a crucial component of positive parenting styles, helps children internalize emotion regulation strategies and develop the emotional skills necessary for navigating social environments. Consequently, emotionally socialized students may be more capable of establishing meaningful social bonds at school ([57]).

When children form emotional ties with and experience a sense of belonging in their educational environment, they are more willing to seek academic support from teachers and peers when needed ([54]) and more likely to approach academic tasks with enthusiasm, motivation, and persistence while demonstrating a lesser tendency to procrastinate ([80]; [82]). That said, the indirect pathway from perceived positive parenting to academic procrastination through school emotional engagement appears limited in explanatory strength. One reason may be that children’s emotional engagement with school is more strongly shaped by proximal in-school relationships, such as peer affiliations and teacher–student interactions, than by parental influences ([39]). While parenting may set the emotional foundation, its influence on school-based emotional connections could be relatively distal compared with direct interpersonal experiences within the school environment. Additionally, the non-significant relationship between perceived parenting style and school emotional engagement suggests that additional explanatory mechanisms may be at play. Prior studies identified parental educational involvement ([15]), students’ grit ([52]), and adaptive coping ([66]) as contributors to school engagement. Future research may benefit from further examining these external and internal influences to uncover the mechanisms through which parenting affects school emotional engagement.

### 4.4. The Chain Mediating Role of Emotional Resilience and School Emotional Engagement

Likewise, the chain mediation effect involving emotional resilience and school emotional engagement was significant yet relatively small in magnitude. The post hoc power analysis suggests that this pathway is reliable and unlikely to be a result of chance. The finding provides tentative support that perceived positive parenting style may influence children’s academic procrastination through a sequence of emotional resilience and school emotional engagement.

The mediation segment of this pathway aligns with the prior studies (e.g., [22]; [56]) reporting a positive correlation between emotional resilience and school emotional engagement. Emotional resilience could help children generate or sustain positive emotions that contribute to greater school satisfaction and classroom engagement, while also buffering adverse impacts of interpersonal violence ([1]; [7]; [66]; [91]). The full pathway could be better understood through the lens of the Broaden-and-Build Theory of Positive Emotions, which posits that enduring positive emotional experiences build psychological resources. In this context, positive parenting may initiate the process by providing emotionally supportive experiences, which over time contribute to the development of emotional resilience. This internal capacity may then help children navigate peer and teacher relationships more effectively, thereby enhancing emotional engagement. Ultimately, strengthened emotional ties to the school environment increase the intrinsic motivation for learning and discourage reliance on avoidance strategies like procrastination ([86]).

While this chain mediation model has theoretical and empirical support, it would be premature to draw broad developmental conclusions based solely on this mechanism, considering its effect size. It is inferred that emotional resilience and school emotional engagement, while relevant, represent only a part of a broader set of emotional influences. Future research should consider alternative emotional pathways, as well as other mechanisms (e.g., cognitive processing, motivational, and/or contextual) that enhance our understanding of how parenting shapes academic procrastination in youth.

### 4.5. Limitations

The current study contributes to the literature by providing initial evidence of emotional mechanisms (i.e., emotional resilience and school emotional engagement) through which perceived positive parenting style affects children’s academic procrastination. Yet, some limitations should be acknowledged. First, although all mediation effects reached statistical significance, their magnitudes were relatively modest. As such, these findings should be interpreted with caution and regarded as preliminary rather than conclusive. The current results may be partly attributed to the model controlling for baseline levels of academic procrastination, emotional resilience, and school emotional engagement at Time 1. While this approach enhanced the statistical rigor by capturing within-person change, it may have simultaneously attenuated the magnitude of the effects. Future studies could build on this by further evaluating the robustness of emotional mechanisms and exploring mediators of other developmental domains. Second, the data in this study were derived from self-report questionnaires completed by children, which may introduce response bias. Related, some items in the scales (e.g., academic procrastination) may have required a level of reading comprehension that exceeded the participants’ capacities. Future studies should take this into account by selecting measures that are developmentally appropriate and providing adequate guidance to ensure accurate interpretation of the items. In addition, although the parenting style scale has consistently demonstrated good measurement properties in previous studies ([70]; [65]), the overprotection dimension in the current study yielded relatively low reliability (0.549 for fathers and 0.565 for mothers), possibly due to the sample characteristics, and was, therefore, removed from the analysis to ensure data quality. This exclusion may have resulted in a less comprehensive characterization of positive parenting style and may limit the ability to fully capture its influence on academic procrastination. Finally, the current study conceptualized emotional resilience as a precursor of school emotional engagement. While this direction is theoretically and empirically supported, a reversed or reciprocal relationship is possible. Given that school, like family, also functions as a proximal microsystem for children’s development, future research could consider employing methods such as cross-lagged analysis to assess bidirectional influences between emotional resilience and school engagement.

### 4.6. Practical Implications

As this study revealed the emotional mechanisms through which positive parenting influences children’s academic procrastination, it is important to support parents in strengthening their own emotional functioning. Doing so enables them to serve both as role models of emotional regulation and recovery and as sources of emotional support for their children ([12]).

By examining children’s emotional resilience in everyday contexts rather than solely in response to major traumas, the current study illustrates that children’s capacity to cope with daily stressors can also be strengthened through the development of resilient emotional and behavioral strategies. Thus, incorporating programs that foster emotional resilience into psychoeducational efforts may represent a promising avenue for supporting academic outcomes in youth.

In addition to engaging parents and children, it is equally crucial to involve schools in supporting children’s academic experiences ([87]). A strategy worth considering for enhancing students’ emotional engagement with school is to embed this goal into school policies and practices. Schools may implement policies that encourage positive peer interactions and supportive teacher–student relationships. This could be achieved through structured school activities (e.g., cooperative learning projects, peer mentoring programs) and through teacher professional development focused on building emotionally supportive classroom climates and fostering relational teaching practices ([3]). In collectivist cultural contexts such as China, where interdependence and relational harmony are emphasized, fostering interpersonal connections could strengthen students’ emotional engagement with school and mitigate the risk of academic procrastination.

## 5. Conclusions

Utilizing three waves of data, the current study revealed that the perceived positive parenting style could, though in a limited capacity, influence academic procrastination in children through emotional resilience and school emotional engagement, both individually and sequentially. In addition to reducing children’s procrastinating behaviors through enhancing their cognitive functioning and dispositional attributes, parents may also need to invest in nurturing children’s emotional strengths. By identifying the emotional correlates of academic procrastination, the present study suggests that both research and practice should recognize and address the emotional dimension when supporting young people’s academic development.

## Figures and Tables

**Figure 1 behavsci-15-00890-f001:**
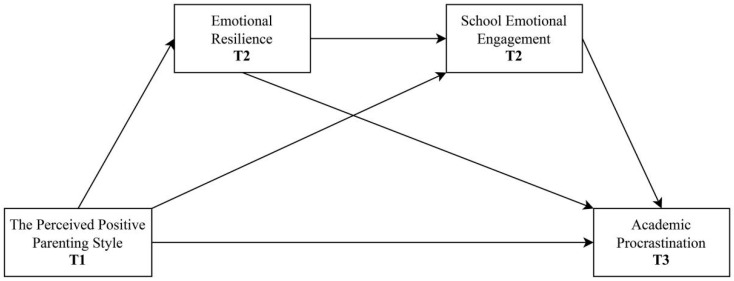
Hypothetical model diagram of the relationship between the perceived positive parenting style and academic procrastination.

**Figure 2 behavsci-15-00890-f002:**
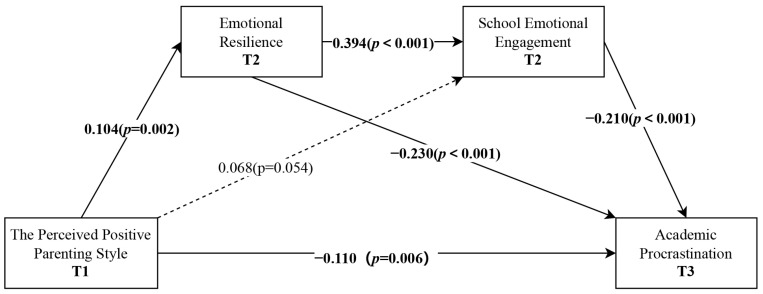
Statistical diagram of structural equation modeling.

**Table 1 behavsci-15-00890-t001:** Demographic characteristics of participants (*N* = 728).

Variable						
Age (*M*, *SD*)	9.84 (0.77)					
Gender	Male (50.8%)	Female (49.2%)				
Economic level	Impoverished (0.1%)	Not Wealthy (5.5%)	Average (69.6%)	Somewhat Wealthy (22.5%)	Wealthy (2.2%)	
Parents’ level of education	Elementary school or below (2.9%)	Middle School (10%)	High School (18%)	College (18.4%)	Bachelor Degree (30.5%)	Graduate Degree (20.2%)

**Table 2 behavsci-15-00890-t002:** The means, standard deviations, and correlations among the variables.

Variable	*M*	*SD*	1	2	3	4	5	6
1. Positive parenting style	3.29	0.52						
2. T1 Emotional Resilience	4.27	0.97	0.517 **					
3. T2 Emotional Resilience	4.21	1.05	0.377 **	0.583 **				
4. T2 School Emotional Engagement	4.19	0.71	0.447 **	0.472 **	0.392 **			
5. T2 School Emotional Engagement	4.11	0.74	0.380 **	0.402 **	0.563 **	0.550 **		
6. T1 Academic Procrastination	2.07	1.53	−0.240 **	−0.243 **	−0.112 **	−0.270 **	−0.172 **	
7. T3 Academic Procrastination	1.87	0.65	−0.323 **	−0.356 **	−0.411 **	−0.315 **	−0.415 **	0.283 **

Note. T1 = Time 1, T2 = Time 2, T3 = Time 3. ** *p* < 0.01.

**Table 3 behavsci-15-00890-t003:** Parameters of the chain mediation model.

	Bias-Corrected Bootstrapped Estimates for the Effects	
	*β*	95%CI
***Direct Pathway*** Positve Parenting Style (T1) → Academic Procrastination (T3)	−0.110	[−0.190, −0.035]
** *Indirect Pathways* **		
IND1: Positve Parenting Style (T1) → Emotional Resilience (T2) → Academic Procrastination (T3)	−0.024	[−0.045, −0.009]
IND2: Positve Parenting Style(T1) → School Emotional Engagement (T2) → Academic Procrastination (T3)	−0.014	[−0.033, −0.002]
IND3: Positve Parenting Style (T1) → Emotional Resilience (T2) → School Emotional Engagement (T2) → Academic Procrastination (T3)	−0.009	[−0.017, −0.003]

## Data Availability

The data presented in this study are available on request from the corresponding author.

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
