# Peer review of "How Perceived Positive Parenting Style Protects Against Academic Procrastination in Children: The Mediating Roles of Emotional Resilience and School Emotional Engagement"

_behavsci, 2025, doi:10.3390/bs15070890_

Round 1
Reviewer 1 Report (New Reviewer)
Comments and Suggestions for Authors
Thank you for the opportunity to review “How Perceived Positive Parenting Style Protects Against Academic Procrastination in Children: The Mediating Roles of Emotional Resilience and School Emotional Engagement”, which was a quantitative study examining the serial mediation effects of positive parenting, emotional resilience, school emotional engagement, and ultimately, academic procrastination. It appears this paper is a revision, though it is my first time reading the paper myself. The authors have clearly put a lot of work into this paper, and I only have a few minimal comments to improve the manuscript. I organize my comments by section:
Intro:
Page 3, line 135; “This study proposes”-The authors should avoid animating inanimate objects. They should say “we propose” if anything.
Page 5, line 200: “Limited is known”-Unclear phrase. I would revise to say “limited information is known” or “little is known” to improve clarity.
Method:
Page 8, line 371: There is a reference to Zhao et al., 2022 in the footnote margins, but I am not sure why it’s there.
Page 10 lines 410-425: The authors do not provide CFA information for the school emotional engagement nor the academic procrastination scales, but they do so for the previous measures. Why not report CFA results for all measures? Citing past psychometric evidence has fallen out of favor in recent years, and reporting results for some scales and not others makes it look like the CFA results were poor for these two scales and the authors are trying to hide it. I recommend the authors include the missing CFA results for the analytic sample to verify the psychometric properties of their measures.
The authors should indicate how the indirect effects were derived in Mplus. There are multiple ways to do it (e.g., Model indirect commands, model constraint commands, etc.). This would be useful to know to ensure the reader that the authors actually computed indirect effects.
Results:
Several times in the results and discussion section, the authors note that the findings for the indirect effect of positive parenting on academic procrastination through school emotional engagement should be interpreted cautiously, which is something I agree with, but they give the rationale that the effect sizes are small. That’s not the reason I would be suspicious of the findings. Instead, I would make note that the effect of positive parenting on school emotional engagement (the a-path) was not significant, meaning that the indirect effect (axb) was based entirely on the b-path, which is weird and not very compelling.
Discussion:
Page 15, line 595: Typo detected: “When children forms” should be “When children form”
Page 16, line 684: Typo detected: “Future studies could build on this by furthering evaluate”-“evaluate” should be “evaluating”
Page 17, line 693: The authors mention that analyses excluded data from the overprotection dimension, but this was not mentioned previously based on my reading. The authors should note this in the methods section, affirming that the results were unaffected by the low reliability of the dimension.
Page 17, line 710: Typp detected: “Prior study suggests” should be “Prior studies suggest”
Overall, this paper is very well-written and contributes meaningfully to the literature. I applaud the authors on their efforts.
Author Response
Thank you for the opportunity to review “How Perceived Positive Parenting Style Protects Against Academic Procrastination in Children: The Mediating Roles of Emotional Resilience and School Emotional Engagement”, which was a quantitative study examining the serial mediation effects of positive parenting, emotional resilience, school emotional engagement, and ultimately, academic procrastination. It appears this paper is a revision, though it is my first time reading the paper myself. The authors have clearly put a lot of work into this paper, and I only have a few minimal comments to improve the manuscript. I organize my comments by section:
Intro:
Page 3, line 135; “This study proposes”-The authors should avoid animating inanimate objects. They should say “we propose” if anything.
Response: Thank you for your suggestion regarding writing improvement. We have revised “This study proposes” to “we propose” accordingly to enhance clarity and readability.
Page 5, line 200: “Limited is known”-Unclear phrase. I would revise to say “limited information is known” or “little is known” to improve clarity.
Response: Thank you for pointing out the grammatical error. We have corrected “limited is known” to “little is known” accordingly.
Method:
Page 8, line 371: There is a reference to Zhao et al., 2022 in the footnote margins, but I am not sure why it’s there.
Response: Thanks for catching this. It may have been due to the formatting. In the revised manuscript, we have removed this footnote.
Page 10 lines 410-425: The authors do not provide CFA information for the school emotional engagement nor the academic procrastination scales, but they do so for the previous measures. Why not report CFA results for all measures? Citing past psychometric evidence has fallen out of favor in recent years, and reporting results for some scales and not others makes it look like the CFA results were poor for these two scales and the authors are trying to hide it. I recommend the authors include the missing CFA results for the analytic sample to verify the psychometric properties of their measures.
Response: Thank you for pointing out this oversight in the manuscript. We have added the CFA results for the School Emotional Engagement and Academic Procrastination scales. Specifically, the CFA for the School Emotional Engagement scale yielded good model fit indices: χ2 / df = 3.59, CFI = 0.984, TLI = 0.958, RMSEA = 0.060. The CFA for Academic Procrastination scale also indicated acceptable fit: χ2 / df = 3.14, CFI = 0.946, TLI = 0.934, RMSEA = 0.047. The results indicate that the structural validity of both scales is acceptable.
The authors should indicate how the indirect effects were derived in Mplus. There are multiple ways to do it (e.g., Model indirect commands, model constraint commands, etc.). This would be useful to know to ensure the reader that the authors actually computed indirect effects.
Response: Thank you for pointing this out. We tested the indirect effects in Mplus using the model constraint command. We also applied bias-corrected bootstrapping with 5,000 resamples to generate 95% confidence intervals for the indirect effects. This information has been added to Section 2.4 “Data Analysis” in the revised manuscript.
Results:
Several times in the results and discussion section, the authors note that the findings for the indirect effect of positive parenting on academic procrastination through school emotional engagement should be interpreted cautiously, which is something I agree with, but they give the rationale that the effect sizes are small. That’s not the reason I would be suspicious of the findings. Instead, I would make note that the effect of positive parenting on school emotional engagement (the a-path) was not significant, meaning that the indirect effect (axb) was based entirely on the b-path, which is weird and not very compelling.
Response:
Thank you for bringing up your concern regarding the indirect effect of positive parenting on academic procrastination via school emotional engagement. You noted that the a-path (from positive parenting to school emotional engagement) was not statistically significant, which caused your doubt about the validity of the mediation effect (a × b).
We acknowledge this concern and appreciate the opportunity to clarify. While the traditional causal steps approach (e.g., Baron & Kenny, 1986) required each path (a and b) to be significant to support a mediation claim, more recent methodological perspectives have shifted away from this piecemeal logic. As noted by Hayes and Rockwood (2017), the significance of the individual paths a and b is not a prerequisite for establishing a valid indirect effect. Instead, what truly matters is whether the product term (a × b) differs from zero by an inferential standard such as a confidence interval or null hypothesis test. Hence, in this case, the non-significant effect of positive parenting on school emotional engagement does not influence the significance and validity of the mediation effect of school emotional engagement.
We have added the content and the citation to highlight that the non-significant individual effect of perceived positive parenting style on school emotional engagement does not necessarily determine the significance of the mediation effect of school emotional engagement.
Baron, R. M., & Kenny, D. A. (1986). The moderator–mediator variable distinction in social psychological research: Conceptual, strategic, and statistical considerations. Journal of personality and social psychology, 51(6), 1173.
Hayes, A. F., & Rockwood, N. J. (2017). Regression-based statistical mediation and moderation analysis in clinical research: Observations, recommendations, and implementation. Behaviour research and therapy, 98, 39-57.
Discussion:
Page 15, line 595: Typo detected: “When children forms” should be “When children form”
Response: Thank you for pointing out the grammatical error. We have corrected “When children forms” to “When children form” accordingly.
Page 16, line 684: Typo detected: “Future studies could build on this by furthering evaluate”-“evaluate” should be “evaluating”
Response: Thank you for pointing out the grammatical error. We have corrected “evaluate” to “evaluating” accordingly.
Page 17, line 693: The authors mention that analyses excluded data from the overprotection dimension, but this was not mentioned previously based on my reading. The authors should note this in the methods section, affirming that the results were unaffected by the low reliability of the dimension.
Response: Thank you for pointing this out. In Section 2.2.1 “Positive Parenting Style”, we have added the rationale for excluding the overprotection subdimension, namely its low Cronbach’s alpha (below 0.6). Additionally, we conducted a CFA on the composite positive parenting style measure, which was derived from the reverse-scored rejection subscale and the emotional warmth subscale. The results indicated that this combined construct demonstrated good structural validity (χ2 / df = 3.59, CFI = 0.960, TLI = 0.941, RMSEA = 0.06).
Page 17, line 710: Typp detected: “Prior study suggests” should be “Prior studies suggest”
Response: Thank you for pointing out the grammatical error. We have revised the wording of this part during the revision process. Specifically, this section has been revised as follows: As this study reveals the emotional mechanisms through which positive parenting influences children’s academic procrastination, it is important to support parents in strengthening their own emotional functioning. Doing so enables them to serve both as role models of emotional regulation and recovery, and as sources of emotional support for their children (Brumariu, 2015).
Overall, this paper is very well-written and contributes meaningfully to the literature. I applaud the authors on their efforts.
Again, thank you for your valuable suggestions and positive assessment of our work.
Reviewer 2 Report (New Reviewer)
Comments and Suggestions for Authors
Dear Authors,
Thank you for allowing me to review your manuscript. Overall, it is an interesting and well researched paper that offers a new perspective to the field. However, I do have serious concerns with the lack of detail in the methods and some of the inferences made. I offer these comments in good faith - please do not be offended as I believe your paper has merit. I would strongly suggest that you expand on what the chain mediation model is and why it was selected. Additional comments can be found on the attached.

Author Response
Reviewer 2
Thank you for allowing me to review your manuscript. Overall, it is an interesting and well researched paper that offers a new perspective to the field. However, I do have serious concerns with the lack of detail in the methods and some of the inferences made. I offer these comments in good faith - please do not be offended as I believe your paper has merit. I would strongly suggest that you expand on what the chain mediation model is and why it was selected. Additional comments can be found on the attached.
Introduction
Line 35: I understand the point you are making, but are children aged 5-8 striving for academic excellence?
Response: Thank you for raising this important point. In the Chinese context, parents tend to place a high value on academic achievement from an early stage. As a result, children—whether actively or passively—are often driven to engage in learning and participate in various extracurricular or enrichment programs in pursuit of academic excellence. Therefore, we believe that exploring the mechanisms influencing Chinese children's academic development is of significant importance for promoting their adaptive growth. To better highlight the cultural background of the participants, we have revised the original sentence to: “Striving for academic success constitutes a central developmental task for children in China.”
Line 55: Suggest adding context around the Broaden-and-Build Theory of Positive Emotions.
Response: In the opening paragraphs of the Introduction, we first introduced the Broaden-and-Build Theory of Positive Emotions as the theoretical framework guiding this study. In the newly uploaded manuscript, we have added an explanation of the core mechanisms underlying the Broaden-and-Build Theory of Positive Emotions in this section.
Further elaboration is provided in Section 1.2, “The Mediating Effect of Emotional Resilience,” where we discuss its relevance to the research objectives. Specifically, we highlighted the role of positive emotions in broadening individuals’ thought–action repertoires and promoting adaptive functioning. We also cited empirical studies supporting the beneficial effects of positive emotions in enhancing academic performance and alleviating academic procrastination.
Line 70-71: All true, but stressors, both perceived and real, will be very different for children and adolescents.
Response: Thank you for your comment. Your point is well taken. Children and adolescents do differ in many aspects, including the types and levels of pressure they experience and perceive. As the participants in our study were children, we have removed the references of adolescents throughout and revised the terminology to consistently refer to the sample as children.
Line 82: Children and adolescents? Assuming a word has been missed.
Response: Thank you for your comment. We have reviewed the manuscript and revised the terminology to consistently refer to the participants as “children,” removing references to “adolescents.”
Line 93: Incorrect reference formatting used (should be &, not and).
Response: Thank you for pointing out the formatting error in our citation. We have corrected “and” to “&” accordingly.
Line 106: You state that ‘… as Chinese society increasingly embraces values such as openness, equality, and democracy, contemporary parent-child relationships have turned more egalitarian (Xia et al., 2015).’ I appreciate that you have used a citation here, but is Chinese society really embracing openness and democracy? Many would suggest the opposite. Is there evidence from a school curriculum perspective?
Response:
Thank you for raising this important point. In the revised manuscript, we have clarified that the use of terms such as “openness” and “democracy” refers specifically to evolving values in the domains of parenting and education, rather than broader political ideologies. These values are reflected in changing parent-child dynamics in China, where more families now emphasise respect, emotional responsiveness, and open communication.The policy document issued by China’s Ministry of Education, Opinions on Improving the Collaborative Education Mechanism Among Schools, Families, and Society (jointly released by thirteen government departments), explicitly emphasizes the scientific parenting principle of showing greater respect and understanding toward children, and strengthening equal communication. Moreover, national curriculum reforms—such as the emphasis on “core competencies” and socio-emotional development—support the trend toward more egalitarian and student-centred educational practices. We have revised the text accordingly to prevent potential misunderstanding.
The original sentence has been revised to: However, as modern educational values such as openness, mutual respect, and emotional communication gain wider acceptance in China, contemporary parent-child relationships have gradually become more egalitarian (Xia et al., 2015).
Policy document link:
http://www.moe.gov.cn/srcsite/A06/s3325/202301/t20230119_1039746.html
Line 123. Limited what? Is there a word missing here?
Response: Thank you for pointing out the grammatical error. We have corrected “limited is known” to “little is known” accordingly.
Line 172: The Chain Mediating Effect of Emotional Resilience and School Emotional Engagement Very informative content but I am left wondering, where is this going? It is reading like a literature review. In this section there is talk of resilience and trauma yet only in the concluding sentence is procrastination mentioned. There should be clearer links if this section is to remain.
Response: Thank you for your feedback. We have revised this section by having introducing the chain mediation pathway at the beginning of the section, adding references to procrastination, and putting related content together to ensure a more coherent and targeted explanation of how emotional resilience and school emotional engagement are linked to jointly account for the effect of parenting style on academic procrastination.
Line 196: Again, consistency is needed. Children and/or adolescents.
Response: Response: Thank you for your comment. We have reviewed the manuscript and revised the terminology to consistently refer to the participants as “children,” removing references to “adolescents.”
Line 194: Current study
Has this method been used by others? A citation may add value here as it’s unclear how you cam about this method.
Response: Thank you for your suggestion. The method we employed is commonly referred to by several names, including chain mediation model, sequential mediation model, or serial multiple mediator model. It has been widely applied in research aiming to examine how multiple mediators function in a causal sequence—where one mediator influences subsequent mediators along the pathway to the outcome. We have referenced a highly cited article (Hayes & Rockwood, 2017) to support the use of this analytic approach.
Methods – Participants
Line 214. There are serious formatting issues here – Table 1 is unreadable.
Response: Thanks for noting this. In the revised manuscript, we have corrected this issue by properly presenting the table containing the demographic information.
Line 215: How were the participants recruited? How was parental consent obtained?
Response: Thank you for pointing out the missing information in the manuscript. First, participants were recruited using a cluster sampling method. We have added the following sentence to Section 2.1 Participants: “Participants were recruited using a cluster sampling method.”
Second, parental consent was obtained by asking students to take the informed consent form home for their parents to sign, and teachers explained the study to parents via online communication prior to the assessment. This has been added to Section 2.3 Procedure as follows: “As primary school students are under the age of 18, they were instructed to take the informed consent form home for parental signature. Teachers also explained the study to parents through online communication prior to the assessment.”
Line 230: There appears to be statistical methods. If these relates to the citations these should be clearly explained as otherwise it looks as though you are including your own statistical methods. Please carefully check before submitting.
Response: Thank you for bringing this up, which helped us clarify this issue. The overprotection dimension was excluded from the analysis due to its low internal consistency. Rejection and emotional warmth are two negatively correlated dimensions. In psychological measurement, it is a common practice to reverse-score one of the opposing dimensions before combining them into a composite construct. However, since subdimensions can also have distinct characteristics, such a combination should be approached with caution. Therefore, we referred to the method used by Kang et al. (2024) as a precedent. After combining the two dimensions, we conducted a confirmatory factor analysis (CFA) to test the validity of the positive parenting construct. The CFA results were χ²/df = 3.58, CFI = 0.960, TLI = 0.941, and RMSEA = 0.06, indicating that the model fit was acceptable and the construct was valid. We have revised the manuscript to explain this part accordingly.
Procedure
Line 267: To confirm, the questionnaires were completed in-person and not online?
Response: Thank you for raising this concern. As stated in the manuscript, “Two trained graduate students were arranged in every classroom to guide and supervise the participants in completing the questionnaires,” the assessment was conducted in person and completed by the participants themselves. To make this point clearer, we have revised the sentence to: “The assessment was conducted in person, and all participants completed the questionnaires on-site under the supervision of two trained graduate students assigned to each classroom.”
Line 286: What is the scale of the effect sizes? When providing effect sizes as a data metric it is common to provide the reader with the effect scale sizes/ranks.
Response: Thank you for your comment. We were not entirely certain what was meant by “the scale of the effect size,” but to the best of our understanding, you may be referring to whether established benchmarks were used to interpret the magnitude of the reported effects. If that is the case, we would like to clarify that there is no universally accepted criterion for evaluating the size of standardized mediation effects. In the current study, for example, the mediation effect of emotional resilience on the relationship between parenting style and academic procrastination was β = –0.024, which is generally considered small. However, when examining the proportion of the total effect this pathway accounts for, emotional resilience mediates approximately 15.3% of the total effect (–0.024/ –0.157)—a value that could be viewed as falling between small and medium, depending on the interpretive framework used. As such, different metrics can lead to differing conclusions about effect size. In this study, we focused on the standardized indirect effect (β) to represent effect magnitude and noted that all mediation effects were relatively small.
We have added the related content under Data analysis to indicate this.
Line 291: Eigenvalues are mentioned but these were not mentioned in the statistical analysis. Given that eigenvalues are normally associated with PCA I would suggest expanding on these in your methods.
Response: thanks for the suggestion. We have expanded on this step under the Data analysis section.
Chain mediation model testing
Line 296-297: You’ve mentioned models and statistical methods that really should be defined in the methods and statistical analysis section as it is becoming confusing in what was assessed and why. The information under this section is informative yet given that it wasn’t included in the methods I am struggling to understand how it fits into the overall picture. Statistical methods such as CI and RMSE are included, yet these should have been clearly explained prior to this.
Response:
Thank you for your comment. In Section 2.4 Data Analysis, we provided information on the analytical procedures used to generate the results. Specifically, we first described the variable structure used in structural equation modeling conducted with Mplus 8.0, including the method for estimating missing data, baseline controls, and the criteria for evaluating model fit. Second, bootstrapping was used to extract confidence intervals (CIs) to test the significance of the mediation effects. Third, to ensure the robustness of the mediation analysis, we additionally conducted a Monte Carlo simulation as a post-hoc power analysis.
The results of the above analyses can be found in Section 3.2 Chain Mediation Model Testing of 3 Results. For the model results, we first reported the model fit indices (including the comparative fit index (CFI), Tucker–Lewis index (TLI), root mean square error of approximation (RMSEA), and standardized root mean square residual (SRMR), with values of CFI and TLI > 0.90, RMSEA < 0.08, and SRMR < 0.08 considered acceptable), followed by the results of the path analysis, including the coefficients and significance levels of each path. Second, we reported the results of the mediation analysis tested using bootstrapping. A mediation effect was considered significant when the confidence interval (CI) did not include zero. These results are also summarized in Table 3. Third, the results of the Monte Carlo simulation are presented in the second paragraph of Section 3.2 Chain Mediation Model Testing. A mediation path was considered to have sufficient post-hoc statistical power when power was equal to or greater than 0.80.
Table 3: Incorrectly formatted. Please amend. Again, CI are mentioned but were not previously included in your methods and analysis.
Response: Thank you for your comment. As mentioned in the previous response, we have added information regarding confidence intervals (CIs) in Section 2.4 Data Analysis. We also reviewed Table 3 and have corrected several formatting errors. Table 3 presents the standardized confidence intervals of the direct path and the three mediation paths, tested using bootstrapping. These results can be found in Section 3.2 “Chain Mediation Model Testing” of the main text.
Discussion
Line 329-331: This reads more akin to a limitation. I would suggest moving this sentence.
Response: thanks for the suggestion. We have relocated the sentence to Limitation.
Line 372-379: The reference to the HPA is important and similarly, that parental warmth has been shown to reduce activity in the amygdala. However, the links to your research are arguably generous as best and I’m unsure what links, if any, can be made. If these are to remain then the links should be stated as speculative at best.
Response: Thanks for the comment. We have added a sentence in the end of the paragraph to acknowledge that these neurobiological processes were not directly examined, and these evidences provide preliminary support for the idea that positive parenting results in children’s emotional resilience.
Line 398: Only in the discussion is attachment mentioned. Are the stronger links between attachment and positive parenting?
Response: Thank you for your comment. We agree that the reference to attachment theory in the discussion section may appear abrupt, as it was not introduced earlier in the manuscript. Given that attachment theory is a well-established framework (certainly, the literature is full of evidencing indicating the link between attachment and positive parenting), we were also concerned that introducing it at this stage might inadvertently overshadow the guiding theoretical framework of our study—the Broaden-and-Build Theory of Positive Emotions—thereby distracting from the intended theoretical focus. To maintain theoretical consistency and avoid confusion, we have removed the sentence referencing attachment theory.
Line 412: I’m not necessarily seeing a strong enough link between relatedness and self-determination theory. This is speculative. A stronger and more plausible links could be related to Maslow, but not necessarily SDT.
Response: Thanks for the comment. We have removed the references of relatedness and self-determination theory, and cited several empirical studies (Chen et al.., 2024; Liu et al., 2025; Reschly et al., 2008) that found parental and individual factors that may mediate the link between parenting and school emotional engagement.
Line 418: Where were the results of the post hoc power analysis?
Response: Thank you for your comment. The results of the Monte Carlo simulation have been reported in the second paragraph of Section 3.2 Chain Mediation Model Testing. The original text is as follows: The results of the Monte Carlo simulation indicate that the mediation effect for emotional resilience (β = -0.024, power = 0.915 > 0.8) and the chained mediation effect (β = -0.009, power = 0.893 > 0.8) are small but significant, and the results are reliable. However, the mediation effect for school emotional engagement (β = -0.014, power = 0.637 < 0.8), despite being statistically significant (95% CI does not contain 0), should be interpreted with caution.
Line 475: ‘You state that … ‘while parenting intervention could potentially benefit from enhancing parental emotional responsiveness, the effectiveness of such programs, particularly in relation to children’s academic performance, requires further validation.’ This is very confusing. If the parent isn’t providing positive parenting, who are the ones intervening? What would such a program look like?
Response: Thank you for pointing out the writing issue in this section. We agree that the original phrasing could easily cause confusion for readers. Therefore, we have revised the wording to focus more clearly on the practical implications of the study, emphasizing the potential benefits of enhancing parental emotional responsiveness for child development. In the context of the Chinese educational system, school–family collaboration programs are increasingly being implemented. These programs often invite parents to schools for parenting skills training. The original text has been revised as follows: As this study reveals the emotional mechanisms through which positive parenting influences children’s academic procrastination, it is important to support parents in strengthening their own emotional functioning. Doing so enables them to serve both as role models of emotional regulation and recovery, and as sources of emotional support for their children (Brumariu, 2015).
Line 488: The ‘school system’ is far more complicated that this. How will community connections be maintained? Is there a school-government/policy- implication?
Response: Thanks for the comment. We agree that the school system is complex and multifaceted. In response, we have revised the discussion section to narrow the focus, emphasizing a more tangible and contextually relevant approach—fostering supportive peer and teacher-student relationships as a means of enhancing school emotional engagement. Given that the notion of “community” functions differently in the China compared to Western societies, we have chosen not to propose implications related to broader community connections. Instead, our emphasis remains on feasible, school-based interpersonal strategies congruent with sociocultural values.
Again, We appreciate your feedback that helps strengthen the quality of our paper.
Reviewer 3 Report (New Reviewer)
Comments and Suggestions for Authors
Please see the attached.

Author Response
Review of Manuscript:
This study investigates a timely and relevant question in developmental and educational psychology: how perceptions of positive parenting styles affect academic procrastination in primary school children, mediated through emotional resilience and school emotional engagement. The manuscript is grounded in well-established theoretical frameworks—namely the Broaden-and-Build Theory and ecological systems theory—and benefits from a three-wave longitudinal design with a large sample (N = 728) of Chinese primary school students. The statistical methods, including structural equation modeling with baseline controls and Monte Carlo simulations, are appropriate and thoughtfully implemented. The internal consistency of the measures and the overall model fit indices are robust, and the authors acknowledge the modest effect sizes with clarity.
That said, I have some concerns about the developmental appropriateness of some items. For instance, the scale item "I often delay tasks that must be completed for my studies" may presume a level of metacognitive reflection or task management awareness that is not uniformly present in 9–10-year-old children. Although strong internal consistency is reported for the procrastination scale (Cronbach’s α = .887 at T3), further evidence on reading comprehension, interviewer assistance, or response validity (especially for any children in the 8-year-old range) would strengthen confidence in the results. Given the sample's described educational background and economic diversity, it is unclear whether any of these children required help in reading or interpreting the questions—an issue not addressed explicitly in the procedure section.
Response:
Thanks for the comment. In the original manuscript (Section 2.3), we indicated that each classroom was staffed with two trained graduate students who assisted participants with any questions about the questionnaire items and ensured standardized administration. Yet, we concur with your concern that some items may have required a level of reading comprehension that exceeded the participants’ capacities. Thus, we have noted this in Section 4.5 Limitation and suggested that future research to select developmentally appropriate scales.
The manuscript contributes novel insight by examining the sequential mediation of emotional resilience and emotional school engagement. However, the mediating effect of school emotional engagement has limited statistical power (0.637), and its interpretation should remain tentative, as the authors appropriately note. The decision to exclude the overprotection subscale from the parenting measure due to low reliability (α < .60) is methodologically justifiable, though it may somewhat limit the comprehensiveness of the parenting style profile. The authors may wish to briefly explore in future revisions whether the exclusion may have introduced any biases.
Response:
We appreciate your observation regarding the potential impact of excluding the overprotection subscale. As noted, the decision was based on the subscale’s low internal consistency (α < .60), which rendered its inclusion psychometrically problematic. Despite this, we agree that its removal may have limited the comprehensiveness of the parenting style profile.
While it is possible that this exclusion introduced some degree of bias, we deem the extent is likely to be limited, given the unreliability of the measure. Assessing the precise nature or magnitude of such bias would require rerunning the full set of analyses with and without the overprotection subscale, which appears challenging at this final stage of revision. That being said, we have noted this issue in the limitation and indicated that the representation of positive parenting in this study is not comprehensive due to the exclusion of the overprotection dimension, which may limit the ability of the present study to fully capture the influence of positive parenting style on academic procrastination.
While the theoretical synthesis is compelling and the statistical execution is sound, the discussion could benefit from tighter integration of cultural critique. The manuscript references Chao (1994) and Smetana (2017) to acknowledge that the authoritarian/authoritative parenting dichotomy may not translate cleanly across EastWest contexts. However, the authors could more explicitly address how parental emotional warmth may be expressed differently in Confucian or collectivist cultures and what this implies for instrument validity and interpretation.
Response:
Thank you for the suggestion. We agree that the expression of parental emotional warmth may differ across cultural contexts, and we appreciate the opportunity to elaborate further. The majority of parents of the children in our sample were born in 1980s and 1990s, a generation more exposed to Western cultural values through education, media, and globalization. While China remains a collectivist society, these younger parents may be more open and direct in expressing emotional warmth through explicit verbal praise, physical affection, and emotional support—behaviors that more closely align with Western conceptualizations of emotional warmth.
From a psychometric perspective, while the positive parenting style scale was originally developed in a Western context, it has been carefully adapted and validated by Chinese scholars, and has been cited over 1,000 times in the China National Knowledge Infrastructure (CNKI) database. This scale is widely used in research on Chinese families and is considered to have good cultural relevance and psychometric properties within the local context. Therefore, we believe the construct measured—perceived parental emotional warmth—reasonably reflects parenting behaviors as experienced by the children in the present study.
We have added relevant content following the interpretation of the direct effect of positive parenting style on academic procrastination (Section 4.1). This addition aims to clarify the cultural shifts in parenting practices and provide context for interpreting the findings within the contemporary Chinese society.
Once again, we sincerely appreciate for all the construct feedback, which has significantly improved the quality of the manuscript.
This manuscript is a resubmission of an earlier submission. The following is a list of the peer review reports and author responses from that submission.
Round 1
Reviewer 1 Report
Comments and Suggestions for Authors
Introduction
The justification of the relationship between emotional resilience and procrastination is not fully substantiated in the second paragraph of the introduction.
It is recommended to include a widely accepted definition of parenting style as recognized by the scientific community. Its omission should be addressed to provide greater consistency to the proposed argument in the article.
Methodology - ***
Line 190 – It is unclear whether this is a path analysis and, more specifically, a chain mediation model. It would be advisable to clearly specify the type of statistical study to avoid confusion. A path analysis is a specific SEM model without latent variables, and it is important to explicitly state this.
Lines 232-235 – The decision to merge two dimensions into a single factor based on the principle of parsimony is not acceptable. This key aspect of the study should be justified, at the very least, through a confirmatory factor analysis that supports the decision to combine these two dimensions. Such an analysis would serve as a preliminary step before testing the proposed hypotheses. If the CFA does not support this fusion, it will be necessary to reconsider the subsequent analyses and the interpretation of the results.
Results
Since it is common practice to include only data related to the study objectives in the results section, it is suggested that the preliminary analysis section (which primarily describes the sample) be moved to the sample description section.
Additionally, in Table 2, the gender variable is being treated as a quantitative variable (M and SD are calculated), raising doubts about the type of correlation used with it. Similarly, economic level and parental education level, which are nominal variables, are also being treated as quantitative.
Discussion
Avoid the confusion between a "longitudinal model" and a multi-wave design, as longitudinal models traditionally measure the same variables at different time points, allowing for the analysis of evolution and even reciprocal relationships (as in the case of cross-lagged panel studies).
The scoping review conducted by González-Brignardello et al. (2024) highlights the scarcity of studies addressing procrastination in children, revealing a gap in the literature on this topic. In this regard, the present study adds significant value by incorporating parenting-related variables and emotional aspects, addressing the need identified in the review. Including this scoping review would enhance the relevance of the study by indicating that it helps fill some of the research gaps in procrastination identified in the review.
Author Response
We highly appreciate your valuable comments. We have made attempts to address the issues you brought up, incorporated your suggestions, and edited throughout the manuscript in the revision. We hope you find the revised manuscript in much better shape and meets your expectation.
1. The justification of the relationship between emotional resilience and procrastination is not fully substantiated in the second paragraph of the introduction.
- Response: Thanks for the comment. We have added related content to introduce the relationship between emotional resilience and academic procrastination in the second paragraph of the introduction.
2. It is recommended to include a widely accepted definition of parenting style as recognized by the scientific community. Its omission should be addressed to provide greater consistency to the proposed argument in the article.
- Response: Thanks for the comment. We have added a discussion on the broadly defined concept of parenting styles in Section 1.1 of the introduction. This section presents a widely accepted definition of parenting style proposed by Baumrind (1967, 1971). We also included Maccoby and Martin’s (1983) addition of neglectful parenting, and Darling and Steinberg’s (1993) updated definition emphasizing emotional and behavioral patterns in childrearing. Furthermore, we have added a discussion on Chinese parenting styles and their impact on child development to provide more comprehensive information.
3. Line 190 – It is unclear whether this is a path analysis and, more specifically, a chain mediation model. It would be advisable to clearly specify the type of statistical study to avoid confusion. A path analysis is a specific SEM model without latent variables, and it is important to explicitly state this.
- Response: Thank you for pointing this out. You are right. Path analysis is a general approach that can model various relationships (direct, indirect, moderation, mediation, etc.) whereas the chain mediation is a specific type of mediation model within path analysis, where mediators are linked in a causal sequence. Our study used the chained mediation model and we have specifically indicated this in the Method. Also, we have also rephrased any places mentioning “path analysis” to avoid confusion.
4. Lines 232-235 – The decision to merge two dimensions into a single factor based on the principle of parsimony is not acceptable. This key aspect of the study should be justified, at the very least, through a confirmatory factor analysis that supports the decision to combine these two dimensions. Such an analysis would serve as a preliminary step before testing the proposed hypotheses. If the CFA does not support this fusion, it will be necessary to reconsider the subsequent analyses and the interpretation of the results.
- Response: We appreciate your informative suggestion on this particular issue. We concur with your notion that parental emotional warmth and rejection can represent the two distinct dimensions, so their combination requires further justification from both statistical and empirical perspectives. Based on your suggestion we conducted a CFA on the positive parenting level, which was synthesized from the emotional warmth dimension and the reversed-scored rejection dimension. The model fit indices (χ2 / df = 3.58, CFI = 0.960, TLI = 0.941, RMSEA = 0.06) indicated a good level of fit, showing that they can load onto a single factor well at least in the current study (it doesn’t necessarily mean they should be treated in this way in other cases). The Cronbach's alpha for this combined variable is 0.862. The specific fit indices and the internal consistency have been added to the Positive Parenting Style section in Measures.Empirically, we found a study (Kang et al., 2024) that adopted a similar approach of combining the rejection and overprotection dimensions to measure negative parenting styles. We have added those evidences to back up the decision of combining the two dimensions into one.
Reference: Kang, Baixue, Yingzhen Li, Xueyan Zhao, Xuenai Cui, Xiaoxuan Qin, Shuang Fang, Jie Chen, and Xiaoyan Liu. "Negative Parenting Style and Depression in Adolescents: A Moderated Mediation of Self-Esteem and Perceived Social Support." Journal of Affective Disorders 345 (2024): 149-56. https://doi.org/10.1016/j.jad.2023.10.132.
The revisions can be found in the Parenting Styles section under Measures.
5. Since it is common practice to include only data related to the study objectives in the results section, it is suggested that the preliminary analysis section (which primarily describes the sample) be moved to the sample description section.
- Response: Thanks for noting this. We have adopted your suggestion and have moved the demographic description table of the sample to the sample description section.
6. Additionally, in Table 2, the gender variable is being treated as a quantitative variable (M and SD are calculated), raising doubts about the type of correlation used with it. Similarly, economic level and parental education level, which are nominal variables, are also being treated as quantitative.
- Response: Thank you for pointing out the issue in data presentation. Based on your suggestion, we have removed these categorical demographic variables from the correlation matrix and only presented the correlation matrix of key variables.
7. Avoid the confusion between a "longitudinal model" and a multi-wave design, as longitudinal models traditionally measure the same variables at different time points, allowing for the analysis of evolution and even reciprocal relationships (as in the case of cross-lagged panel studies).
- Response: Thank you for your correction. Based on your suggestion, the term “longitudinal model” has been replaced with “a multi-wave design” and/or “three waves of data” in the manuscript to avoid confusion.
The scoping review conducted by González-Brignardello et al. (2024) highlights the scarcity of studies addressing procrastination in children, revealing a gap in the literature on this topic. In this regard, the present study adds significant value by incorporating parenting-related variables and emotional aspects, addressing the need identified in the review. Including this scoping review would enhance the relevance of the study by indicating that it helps fill some of the research gaps in procrastination identified in the review.
- Response: Thank you for your positive comments. Your feedback helps improve our paper in the revised version.
Reviewer 2 Report
Comments and Suggestions for Authors
This is an interesting study whic sheds light on how procrastination operates and what are the factors involved in procrastinating behaviours. I'm suggesting some changes and further work which would improve the quality of the manuscript.
Parenting Style and Emotional Resiliense Scales should be checked for validity and not only for reliability. Please, run Confirmatory Factor Analysis and report the results.
Both in the introduction and the discussion section, the Parenting Style is purelly elaborated. Especially the discussion would be improved if more theory and interpretation is presented in the text.
Also, a possibly reciprocal relatiosnhip between resilience and engagement should be reported.
Author Response
This is an interesting study which sheds light on how procrastination operates and what are the factors involved in procrastinating behaviours. I'm suggesting some changes and further work which would improve the quality of the manuscript.
Response: We highly appreciate your thoughtful comments and have made attempts to address the issues you brought up, incorporated your suggestions, and edited throughout the manuscript in the revision. We hope you find the revised manuscript in much better shape and meets your expectation.
1.Parenting Style and Emotional Resilience Scales should be checked for validity and not only for reliability. Please, run Confirmatory Factor Analysis and report the results.
Response: Thanks for pointing out this. According to your suggestion, we conducted a confirmatory factor analysis on the parenting style and emotional resilience scales in the Measures section. The results indicate that both scales exhibit a good level of validity and reliability.
The Cronbach's alpha for positive parenting style is 0.862. The model fit indices of positive parenting level (χ2 / df = 3.58, CFI = 0.960, TLI = 0.941, RMSEA = 0.06) indicated a good level of fit, showing that they can load onto a single factor well at least in the current study (it doesn’t necessarily mean they should be treated in this way in other cases). The model fit indices of emotional resilience (χ2 / df = 3.99, CFI = 0.973, TLI = 0.954, RMSEA = 0.064) indicated a good level of fit.
The specific fit indices have been added to Positive Parenting Style and Emotional Resilience section in Measures.
2.Both in the introduction and the discussion section, the Parenting Style is purelly elaborated. Especially the discussion would be improved if more theory and interpretation is presented in the text.
Response: Thanks for the comment, which helps us realize that the theoretical elaboration on parenting style falls short, particularly in the discussion section. Thus, according to your suggestion, in the introduction, we have added relevant content on the model of parental emotion socialization and link it to the broaden-and-build theory of positive emotions to strengthen the argument of effects of parenting style on children’s emotional resilience.
In the discussion, we have specifically referenced and applied broaden-and-build theory of positive emotions, model of parental emotion socialization, and attachment theory in the interpretation of mediation effects of emotional resilience and school emotional engagement. Additionally, we have added neurophysiological evidence to indicate that positive parenting style (mainly through attachment) can regulate children’s HPA axis response to stress in Section 4.2 of the discussion. We hope you find the narratives on the parenting style in the introduction and discussion section have a better balance involving both personal elaboration and theoretical reasoning.
3.Also, a possibly reciprocal relationship between resilience and engagement should be reported.
Response: Thank you for pointing this out. The reciprocal relationship between emotional resilience and school emotional engagement is indeed worth considering. In Section 1.5 of the Introduction and Section 4.3 of the Discussion, we have provided an explanation of their sequential relationship within the chain mediation model. Emotional resilience is more of an intrinsic psychological trait, whereas school emotional engagement is a relatively more explicit emotional expression. Therefore, the chain mediation model in this study assumes that the influence of parenting on emotional resilience is more direct, while its impact on school emotional engagement operates through the child's intrinsic traits. The studies by de Faria et al. (2023) and Martínez et al. (2023) cited in the manuscript also confirm the influence of emotional resilience on school emotional engagement. This supports the inferred sequential order of the two variables in the chain mediation model. Although there is theoretical and previous empirical support, we concur with you that the resilience and school emotional engagement can affect each other reciprocally instead of unilaterally. Meanwhile, emotional resilience and school emotional engagement were collected at the same time point in this study. Thus, your concern is more justified as, strictly speaking from a statistical perspective, the nature of their relationship in the current study is more correlational rather than causal. We have acknowledged it as a limitation of our study and calls for future research to investigate the possibility of their reciprocal relationships by approaches such as longitudinal cross-lagged analysis.
Again, we are grateful for your feedback, which strengthens the quality of our revised manuscript.
The uploaded manuscript may contain revision marks. You may consider accepting all revisions for a clearer version.
Round 2
Reviewer 1 Report
Comments and Suggestions for Authors
Procedure – Instruments
It would be advisable to clarify that School Emotional Engagement does not directly measure school emotional commitment. It is important to define the meaning of its two dimensions (school liking and school avoidance) and explain how they may ultimately indicate school engagement.
Results
-
Figure 2 (results of the path analysis) should be placed in the Results section rather than at the end of the Data Analysis section.
3.1. Preliminary Analysis
-
The variables gender, socioeconomic level, and parents' educational level are nominal variables, which are already described in Table 1. However, Table 2 presents categorical or nominal variables again,
-
In the previous review, I suggested that descriptive statistics should be included in the Methods section, specifically in the sample description, since this information is not used to answer any research objectives. While including descriptive values and correlations of the study’s main variables in the Preliminary Analysis section makes sense, the descriptive statistics of the sample variables (e.g., gender, parents’ educational level, and socioeconomic status) should remain in the sample description, as they are not part of the core analyses.
This adjustment will undoubtedly improve the clarity of the study’s objectives and results.
Author Response
Procedure – Instruments
It would be advisable to clarify that School Emotional Engagement does not directly measure school emotional commitment. It is important to define the meaning of its two dimensions (school liking and school avoidance) and explain how they may ultimately indicate school engagement.
Response: Thank you very much for your comments. We have added the definitions of the two dimensions of the School Liking and School Avoidance Questionnaire (school liking and school avoidance) in Section 2.2.3 School Emotional Engagement. We have also clarified that this scale measures students' emotional attitudes toward school rather than their commitment to engagement. This scale has been validated for measuring children's emotional engagement in school (Liu et al., 2024).
Results
Figure 2 (results of the path analysis) should be placed in the Results section rather than at the end of the Data Analysis section.
Response: Thank you for pointing this out. We have moved Figure 2 to Section 3.2 of the Results.
3.1. Preliminary Analysis
The variables gender, socioeconomic level, and parents' educational level are nominal variables, which are already described in Table 1. However, Table 2 presents categorical or nominal variables again.
In the previous review, I suggested that descriptive statistics should be included in the Methods section, specifically in the sample description, since this information is not used to answer any research objectives. While including descriptive values and correlations of the study’s main variables in the Preliminary Analysis section makes sense, the descriptive statistics of the sample variables (e.g., gender, parents’ educational level, and socioeconomic status) should remain in the sample description, as they are not part of the core analyses.
This adjustment will undoubtedly improve the clarity of the study’s objectives and results.
Response: Thank you for pointing this out. We have moved Table 1 to the Participants section in Method 2.1. We have also updated Table 2 to display the descriptive statistics of key variables and the correlation matrix.
Once again, we sincerely appreciate your support of our research. Your valuable feedback has played a crucial role in enhancing the quality of our manuscript.
